# Hydrogen Plasma Treatment of Aligned Multi-Walled Carbon Nanotube Arrays for Improvement of Field Emission Properties

**DOI:** 10.3390/ma13194420

**Published:** 2020-10-04

**Authors:** Dmitriy V. Gorodetskiy, Artem V. Gusel’nikov, Alexander G. Kurenya, Dmitry A. Smirnov, Lyubov G. Bulusheva, Alexander V. Okotrub

**Affiliations:** 1Nikolaev Institute of Inorganic Chemistry SB RAS, 630090 Novosibirsk, Russia; gusel@niic.nsc.ru (A.V.G.); kurenyaag@niic.sbras.ru (A.G.K.); bul@niic.nsc.ru (L.G.B.); 2Institute of Solid State and Material Physics, Technical University of Dresden, 01062 Dresden, Germany; infostelle@tu-dresden.de; 3Laboratory for Terahertz Research, Tomsk State University, 36 Lenin Ave., 634050 Tomsk, Russia

**Keywords:** carbon nanotube arrays, hydrogen plasma treatment, electronic structure, field emission characteristics

## Abstract

Vertically aligned carbon nanotube (CNT) arrays show potential for the development of planar low-voltage emission cathodes. The characteristics of cathodes can be improved by modifying their surface, e.g., by hydrogen plasma treatment, as was performed in this work. The surface of multi-walled CNT arrays grown on silicon substrates from toluene and ferrocene using catalytic chemical vapor deposition was treated in a high-pressure (~10^4^ Pa) microwave reactor. The structure, composition, and current-voltage characteristics of the arrays were studied before and after hydrogen plasma treatment at various power values and durations. CNT tips were destroyed and catalytic iron was released from the CNT channels. The etching rate was influenced by iron particles that formed on the array surface. The lower emission threshold in the plasma-treated arrays than in the initial sample is explained by the amplification factor of the local electric field increasing due to graphene structures of unfolded nanotube layers that formed at the CNT tips.

## 1. Introduction

Due to the high aspect ratio, mechanical strength, chemical inertness, and high electrical conductivity of carbon nanotubes (CNTs), this material is attractive for electronics, electrochemistry, and for the design of composite materials. The applied external electric field is locally amplified near CNT tips to decrease the macroscopic threshold of electron emission into a vacuum [1,2,3]. This property of CNTs can be used to develop low-voltage-emission cathodes for microwave amplifiers, microfocus X-ray tubes, flat panel displays, etc. [4,5,6,7]. The devices can use both individual nanotubes and CNT arrays that are aligned vertically to the cathode surface. The use of CNT arrays allows the emitter power to increase, consequently expanding their range of application. Cathode emission characteristics can be improved by chemically modifying the CNT’s surface, e.g., by depositing nanoparticles and compounds [8,9,10]. Efficient operation of cathodes composed of vertically-aligned CNTs requires the array surface to be geometrically uniform to provide a constant amplification factor and a constant work function at different parts of the surface.

CNT arrays have been successfully prepared using catalytic chemical vapor deposition (CCVD). The structure of CNT arrays depends on a large number of physical and chemical parameters during synthesis [11,12,13,14,15]. Even small deviations in the synthesis parameters from their optimal values lead to defective CNTs and to the formation of amorphous carbon [16], the quantity of which can increase during synthesis [17]. The deposition of reaction products on the CNT surface can significantly affect the cathode emission characteristics [1,3,18]. One possible method to remove the amorphous carbon from the sample surface is by using hydrogen plasma treatment. Plasma interacts effectively with CNT tips so that they are etched and the CNTs are shortened as a result of treatment [19,20,21,22,23,24,25,26,27].

Hydrogen plasma modification of CNT arrays was performed in a number of works [23,24,25,26]. Plasma treatment was used to uncover nickel particles near the CNT tips. In the presence of carbon source, these particles catalyzed the growth of new CNTs and led to the formation of a hierarchical tree-like array structure [28]. It was shown that hydrogen plasma treatment increases the sharpness of CNTs and, therefore, their efficiency when used as the probe tip of the cantilevers in atomic force microscopes [29]. Hydrogen plasma treatment of CNT surfaces was shown to enhance their adhesion to various materials [30], which can be used in the design of dry adhesives [31].

Hydrogen plasma treatment is also used for CNT synthesis, e.g., for the regeneration of the catalyst deposited on the substrate [32]. It was shown that full-cycle CNT growth in hydrogen plasma is possible in the presence of carbon [33,34]. The synthesis of vertically aligned CNT arrays is also possible [35]. The growth of CNT arrays using plasma-enhanced chemical vapor deposition (PECVD) with a subsequent hydrogen plasma treatment to improve cathode’s current-voltage characteristics was studied in [36].

In the present work, planar low-voltage emission cathodes were prepared using the CCVD synthesis of CNT arrays aligned vertically to the substrate surface. We attempted to identify the factors determining the changes in the array structure as a result of post-synthesis hydrogen plasma treatment and to understand how this treatment improves field emission characteristics. Understanding the processes and the mechanism of hydrogen plasma’s effect during the modification of CNT arrays can make it possible to obtain more efficient field emission cathodes with a lower turn-on threshold and increased stability, reliability, and current density, which will expand their range of application.

## 2. Materials and Methods

### 2.1. Synthesis

The CNT arrays were grown on KEF 4,5 (phosphorus-doped) silicon wafers (orientation [100], size 10 × 10 mm^2^) [37,38] in a horizontal quartz reactor. The reactor with the substrates in its center was evacuated to a pressure of ~10^−2^ Pa, then filled with an inert gas (Ar, purity 99.999%) to atmospheric pressure and heated to 800 °C. The reaction mixture (2% ferrocene (purity 99.99%) solution in toluene with a purity of 99.9999%) was injected in the reactor and aerosol was supplied by Ar with a flow rate of 450 mL/min to the synthesis zone. As a result of ferrocene decomposition, a layer of iron nanoparticles, catalysts of CNT growth, was formed on the substrate surface. A continuous supply of the reaction mixture for 1.5 h led to the formation of a vertically aligned CNT array on the substrate surface. The iron content in the CNT array was ~3.5 wt % [39].

The plasma treatment of CNT arrays was conducted in an AX5250M PECVD reactor (Seki Technotron, Tokyo, Japan) with a microwave plasma power up to ~5 kW [40]. The working pressure in the chamber was ~8 × 10^3^ Pa. Hydrogen was used as the plasma-forming gas, which was supplied at a rate of 500 mL/min. In the course of plasma treatment, the power of the microwave generator varied from 600 to 3000 W. The substrate temperature under these conditions ranged from 850 to 1050 °C.

### 2.2. Characterization

The initial CNT arrays and plasma-treated samples were studied by Raman spectroscopy (LabRAM HR Evolution spectrometer, HORIBA Scientific, Tokyo, Japan) using an Ar^+^ laser at 514 nm with a radiation power of 1 mW and a laser beam spot of 1–3 µm, scanning electron microscopy (SEM, DUAL-BEAM FIB/FEI HELIOS 450S microscope, FEI Company, Hillsboro, USA), and transmission electron microscopy (TEM, JEOL-2010 microscope, JEOL, Tokyo, Japan). The composition of the surface and the electronic structure of plasma-treated samples were studied by X-ray photoelectron spectroscopy (XPS) and near-edge X-ray absorption fine structure (NEXAFS) spectroscopy on the Russian–German Beamline at the BESSY II synchrotron radiation facility (Helmholtz-Zentrum Berlin für Materialien und Energie, Berlin, Germany). The pressure of residual gases in the spectrometer’s analytical chamber did not exceed 10^−8^ Pa. The XPS spectra were excited by photons with excitation energy of 830 eV. The energy scale was calibrated with respect to the Au 4f_7/2_ line at 84.0 eV. Background subtraction by the Shirley method [41] and spectra fitting by Gaussian–Lorentz functions were performed using the Casa XPS program (version 2.3.15, Casa Software Ltd, UK). An asymmetric Doniach-Sunjic function was also used for the fitting of C 1s spectra [42]. The NEXAFS spectra were measured in Auger electron yield (AEY) mode. 

The field emission properties of the samples were studied in a vacuum unit as described in detail in [18]. The substrate with a CNT array was placed on a movable arm (cathode) in a chamber evacuated to 10^−3^ Pa. A sawtooth voltage was generated on a flat anode; the measurements were carried out at an amplitude of 0 to 1500 V.

## 3. Results

A specific feature of the proposed CNT array synthesis is that the reactor is constantly fed with the reaction mixture, namely, a solution of ferrocene in toluene. Since the CNT growth starts from the substrate, the excess iron formed on the substrate as a result of ferrocene decomposition throughout the synthesis is captured inside the channels of the nanotubes being formed [43]. Since the catalyst supply and the rate of CNT growth were not completely coordinated, the side faces of the initial arrays showed light stripes formed by the layers enriched with iron nanoparticles [38,43]. About 1760 µm thick vertically aligned CNT arrays (Figure 1a) prepared using the CCVD method were placed on a horizontal molybdenum support and subjected to hydrogen plasma treatment for a certain period of time. The plasma treatment was periodically terminated to control the thickness and morphology of the array. The thickness of the CNT array was determined as the average of five measurements performed at different points. The interval between the measurements was 5–10 min at the initial stages and then increased gradually. The total time of hydrogen plasma treatment was 6 h; as a result, the array thickness diminished almost by 300 µm to 1470 µm (Figure 1b). The etching of the CNT array occurred simultaneously on the sides and on the surface of the sample (Figure 1b,c). The positions and the number of light-colored layers formed on the side of the plasma-treated sample roughly corresponded to those in the layers of the initial sample enriched with iron nanoparticles. These layers exhibited uneven lateral plasma etching (Figure 1b).

In the course of a 600 W hydrogen plasma treatment, the thickness of the CNT array changed in an obviously nonlinear way, as seen in the corresponding curve (Figure 1d). The intervals on the curve show the spread in the array height values over the sample and the measurement error (~5 %). The most intensive decrease in the height of the CNT array occurred for the first 60 min. The etching rate then decreased considerably and remained approximately constant at a value of 30 µm/h. We estimated that 100 µm etching of nanotubes released a sufficient amount of iron to cover the surface with a continuous ~4 nm thick layer. Figure 1c shows that the agglomerates consist of tree-like chains of iron nanoparticles with a size smaller than 100 nm. The linear parts of the chains are formed by the assembled iron nanoparticles released during the etching. Even though the layer of iron particles is not continuous, its presence increased surface conductivity and decreased the penetration of plasma inside the array so that the etching rate of carbon walls decreased. The chemical reactions of hydrogen plasma with carbon materials associated with the formation of volatile compounds have frequently been discussed [44,45]. The gas phase of the reactor is supplied by light hydrocarbon molecules such as CH_4_, C_2_H_2_, C_2_H_4_, C_2_H_6_, etc. formed by the reaction of CNTs with hydrogen plasma [46,47].

We studied how the structure of a 300 µm thick CNT array changed as the power of the hydrogen plasma increased up to 1500 and 3000 W. The treatment was performed for 15 min. The power increase to 1500 W led to the formation of agglomerated species on the array’s surface. The agglomerates consisted of a number of 0.5–1 µm grains formed of iron nanoparticles coated with a thin carbon layer (Figure 2a). As the power (and, consequently, the sample temperature) increased up to 3000 W, individual iron nanoparticles on the surface of the CNT array were sintered into ~10 µm balls (Figure 2b,c). During the course of cooling, the surface of the balls exhibited a structure of symmetrically arranged hollows.

The balls were single crystals of α-Fe coated with a ~20 nm thick graphite layer. Therefore, increasing the plasma power and the sample temperature led to the etching of carbon layers so that iron nanoparticles were isolated out of the CNT channels and formed agglomerates upon further treatment or sinter into micron-sized iron balls under high power. When cooled, the surface of the balls showed a thin graphite layer that formed during the cooling of iron with dissolved carbon. According to X-ray diffraction data, the balls were single crystals.

The surface of the CNT arrays for field emission measurements was modified using a 600 W plasma discharge to prevent fast CNT etching. The SEM data indicated that the CNT walls were deformed and the CNT tips were sharpened (Figure 3a). The average diameter of plasma-treated CNTs was ~50 nm, whereas the average diameter of nanotubes in the initial array was 30–40 nm. Under the action of gas discharge, amorphous and defective carbon was etched. At the same time, individual graphene layers peeled off so that the CNT diameter increased. These layers can be seen in the TEM images of CNTs (Figure 3b). The CNT tips were etched and the cylindrical structure of outer CNT walls was destroyed with the formation of exfoliated graphene flakes attached to the nanotubes’ surface. No encapsulated iron nanoparticles were observed near the CNT tips (Figure 3c).

The elemental composition of the surfaces of the CNT arrays before and after hydrogen plasma treatment was determined from XPS spectra. Analysis of the spectra indicated that the oxygen content increased from ~1 at % in the initial sample to ~10 at % in the plasma-treated sample. The increased CNT defectiveness in the treated sample led to the additional attachment of oxygen-containing groups after a sample was taken in the air and transported in the XPS spectrometer. The higher intensity of the Fe 3p line in the spectrum of the treated sample is explained by the iron encapsulated inside the CNT channels escaping on the array surface (~3 at %) when the CNT tips were destroyed. 

Figure 4 presents the XPS spectra of carbon, oxygen, and iron in the samples before and after hydrogen plasma treatment. The deconvolution of C 1s spectra (Figure 4a) indicates the presence of graphite-like sp^2^ carbon (component at 284.5 eV), disordered regions in the hexagonal carbon network (C_dis_ component at 285.1 eV) [48], carbon in oxygen-containing groups C–O, C=O (286.6 eV), and COOH (288.5 eV), and π plasmon at 290.5 eV [49,50,51,52]. The plasma-treated sample shows the higher intensity of the peaks from C_dis_ and the oxygen-containing groups, indicative of a larger number of defect states on the CNT surface [53]. The XPS O 1s spectra (Figure 4b) indicate the presence of the oxygen-containing C=O and C–O groups (531.8 eV and 533.4 eV, respectively) in the initial and plasma-treated samples. In the latter case, the total intensity of the oxygen spectrum increased by six-fold. The line with the peak at 530.1 eV (corresponding to oxidized iron Me–O) was observed only in the plasma-treated sample. The electronic state of iron was studied using Fe 3p XPS spectra (Figure 4c). After hydrogen plasma treatment, the intensity of the iron line increased by a factor of 20. The spectra also contained oxidized states of di- and trivalent iron with peaks at 55.7 and 57.8 eV, respectively, and metallic iron with a peak at 53.6 eV [54]. The O 1s and Fe 3p spectra indicate an increased amount of oxidized iron in the plasma-treated samples, which confirms the formation of iron nanoparticle agglomerates on the surface. The oxidation of the iron nanoparticle surface resulted from storage of the samples in air. 

Raman scattering spectra of the CNT array samples before and after hydrogen plasma treatment are compared in Figure 5a. The spectra were normalized to the peak G intensity. The spectrum of the initial sample shows peaks D, G, and 2D that are characteristic of multi-walled CNTs [55]. The intensity of peak D significantly increased for the plasma-treated sample, thus indicating increased defect density in the CNTs. The integrated intensity of 2D peaks is comparable to that of peak G, both for the initial and the plasma-treated sample. Since the relative integrated intensity of peak 2D did not change in the spectrum of the plasma-treated sample, we can explain the significant increase in peak D intensity by the edge states in exfoliated graphene flakes.

Figure 5b shows NEXAFS C K-edge spectra of the samples before and after hydrogen plasma treatment. The spectra were normalized to the intensity at 310 eV. The resonances at ~285.6 and ~291.7 eV correspond to electron transitions from 1s levels to π* and σ* states, respectively [56,57]. The peaks between ~287 and ~291 eV refer to the transitions involving carbon atoms bonded to functional groups and to defects [58,59]. The appearance of a spectral feature at ~288.6 eV in the spectrum of the plasma-treated sample indicated the presence of oxidized carbon states. The relative intensity of the π* resonance decreased for that sample. This change could be caused by the changes in the orientation of graphene layers with respect to the incident X-ray beam [60]. It was shown earlier that the ratio of π* and σ* resonances in C K-edge spectra of aligned CNTs depends on the angle of the incident radiation with respect to the surface of the array of multi-walled CNTs [61,62,63,64,65] and single-walled CNTs [60]. The π*/σ* ratio is maximal when the X-ray beam is normal to the sample surface. The plasma-treated samples displayed exfoliated graphene flakes whose orientation relative to the sample surface differed from the tubes orientation; as a result, the angular dependence of the photoelectrons emitted from the sample changed [62,66]. The difference between carbon states in initial and plasma-treated arrays is also demonstrated by the difference between the widths of the π* resonances, which was notably narrower in plasma-treated samples (Figure 5b).

The current–voltage (I–V) characteristics of cathodes from CNT arrays before and after hydrogen plasma treatment are compared in Figure 6. A notable emission current of 0.1 µA from the 1 cm^2^ cathode appears at a field strength of ~0.8 V/µm in the plasma-treated sample. The corresponding value in the initial sample is ~1 V/µm. However, CNT arrays prepared on a copper substrate [67] exhibited a lower emission threshold of ~0.6 V/µm, since the field enhancement factor was larger in this case due to thinner CNTs being synthesized by laser-activated CCVD.

The lower field emission threshold in the plasma-treated samples is explained by the pyrolytic carbon deposit stripping out of the CNT surface while the electron density at the Fermi level increased due to the edge effects of graphene nanoparticles that peeled off the multi-walled nanotubes. The formation of graphene flakes around the CNT tips increased the field enhancement factor (Figure 6). The I–V curve of the plasma-treated sample has a fracture point at ~1.3 V/µm, which is atypical of the tunneling mechanism of electrons. This effect can be attributed to the high mechanical mobility of exfoliated graphene flakes in an electric field. An additional amplification was due to these flakes turning along the field toward the anode [68].

When plotted in Fowler-Nordheim coordinates, the I–V characteristics of the samples are approximately linear in the region of low electric fields (Figure 6b). However, the slopes of the curves are different, which means that either the field enhancement factor increased or the electron work function decreased in the plasma-treated sample. Possibly, both of these factors occurred. It is commonly thought that the work function in multi-walled CNTs is in the interval of 4.7–4.9 eV [69,70]. The work function decrease from 4.8 to 4.5 eV may be due to graphene flakes undergoing doping under an applied electric field [71].

## 4. Conclusions

Multi-walled CNT arrays prepared on silicon substrates by the CCVD method were treated by high-pressure hydrogen plasma at ~8 × 10^3^ Pa with the microwave generator operating at 600, 1500, and 3000 W. The surface structure of the CNT arrays was found to be dependent on the plasma power. At a power of ~3000 W, the CNTs were etched rapidly and iron micron-sized particles formed on the array’s surface. At low power, iron nanoparticles coated the array surface and decreased the etching rate. In this case, the length of the array decreased nonlinearly with time. A detailed analysis of the TEM images showed that the CNT tips were etched, the cylindrical structure of outer walls was destroyed, and exfoliated graphene flakes were formed on the CNT surface. According to the XPS data, the number of surface oxygen groups and defect states increased in the plasma-treated samples. The field emission properties are improved as a result of partial etching of the amorphous deposits formed on the CNT surface during CCVD synthesis, destruction of the CNT tips, formation of graphene flakes, and partial CNT functionalization. Plasma-treated CNT arrays show a lower emission threshold of 0.8 V/µm and a higher current density. This result proves that the proposed hydrogen plasma treatment of CNT arrays can be successfully applied to prepare effective planar field electron emitters.

## Figures and Tables

**Figure 1 materials-13-04420-f001:**
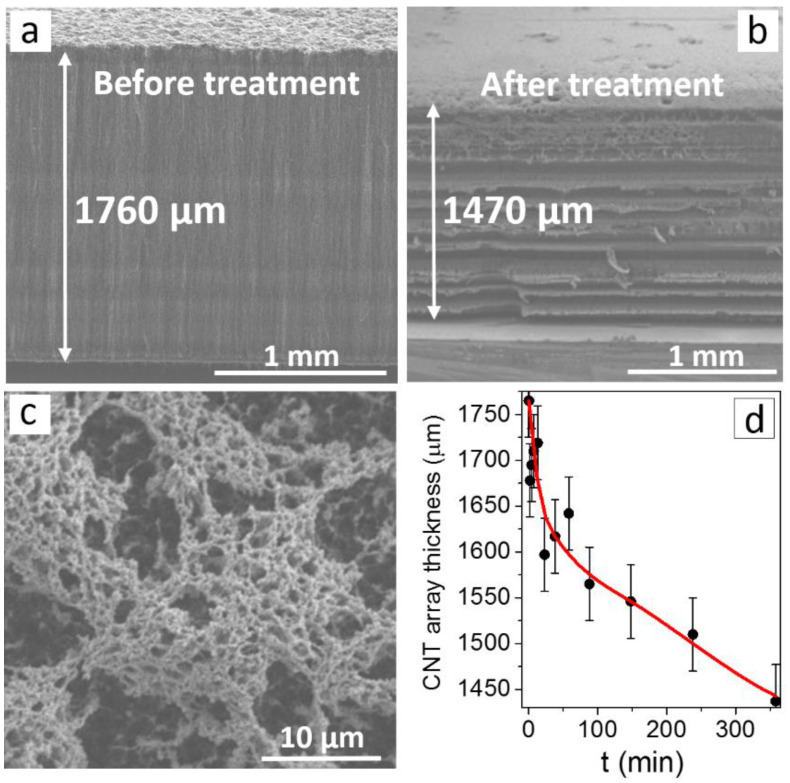
SEM images of the side surface (**a**,**b**) and top surface (**c**) of the carbon nanotube (CNT) array before (**a**) and after hydrogen plasma treatment at 600 W (**b**,**c**). Dependence of the CNT array’s thickness on the hydrogen plasma treatment time (**d**).

**Figure 2 materials-13-04420-f002:**
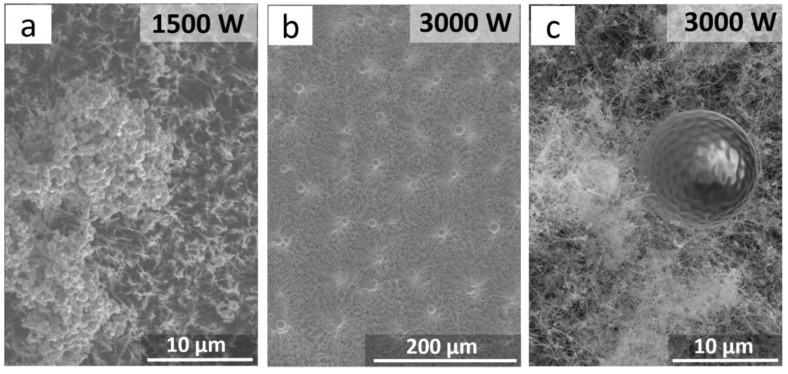
SEM images of the CNT array surface after hydrogen plasma treatment at 1500 W (**a**) and 3000 W (**b**,**c**), showing the formation of iron particles.

**Figure 3 materials-13-04420-f003:**
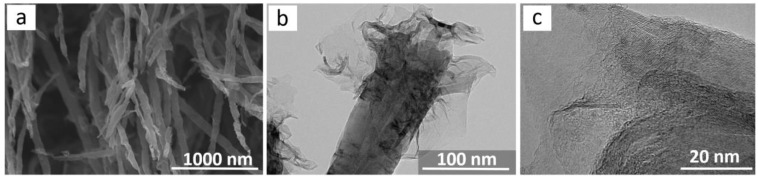
SEM (**a**) and TEM (**b**,**c**) images of CNTs after the array was treated with hydrogen plasma.

**Figure 4 materials-13-04420-f004:**
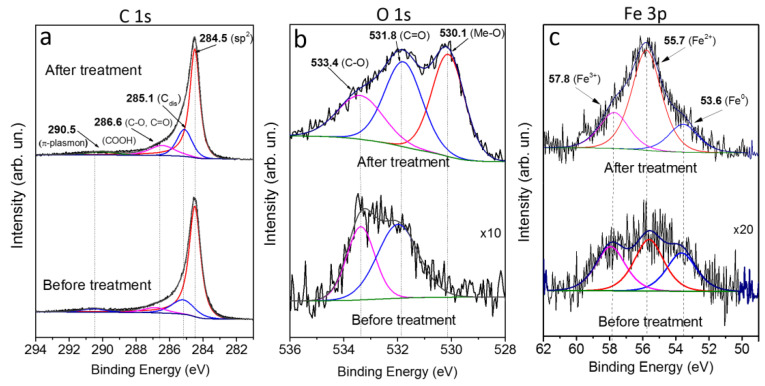
C 1s (**a**), O 1s (**b**), Fe 3p (**c**) XPS spectra of the initial (bottom) and the plasma-treated (top) CNT samples.

**Figure 5 materials-13-04420-f005:**
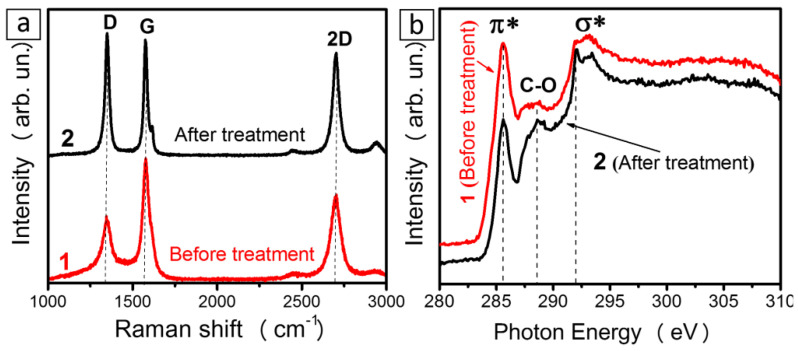
Raman spectra (**a**) and near-edge X-ray absorption fine structure (NEXAFS) C K-edge spectra (**b**) of CNT arrays before (1) and after (2) hydrogen plasma treatment.

**Figure 6 materials-13-04420-f006:**
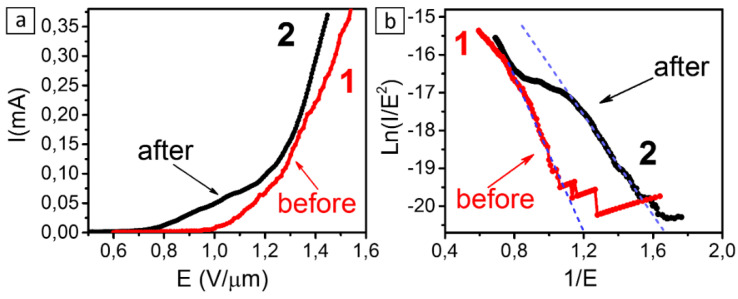
Current–voltage (I–V) characteristics measured for the initial and plasma-treated CNT arrays (**a**). The curves plotted in the Fowler-Nordheim coordinates (**b**).

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
