# Peer review of "Hydrogen Plasma Treatment of Aligned Multi-Walled Carbon Nanotube Arrays for Improvement of Field Emission Properties"

_materials, 2020, doi:10.3390/ma13194420_

Round 1
Reviewer 1 Report
Thank you for presenting your work to Materials.
I have reviewed the manuscript but have come comments which are required to addressed carefully before manuscript can be deemed publishable.
Please see the attached pdf for my comments.
Please do a point by point rebuttal and accordingly revise the manuscript.

Author Response
We thank the Reviewers for the interest to our work and for useful comments. Bellow we provide point-to-point responses and describe the changes made in the manuscript according to the comments.
Reviewer #1: Recommendation:
Thank you for presenting your work to Materials. I have reviewed the manuscript but have come comments which are required to addressed carefully before manuscript can be deemed publishable. Please see the attached pdf for my comments. Please do a point by point rebuttal and accordingly revise the manuscript.
Comments
- Novelty? Please add up few lines here to better explain the novelty of your work here. (line 62)
Authors’ response:
The following additional paragraph has been added to the manuscript: “Understanding the processes and the mechanism of hydrogen plasma's effect during the modification of CNT arrays can make it possible to obtain more efficient field emission cathodes with a lower turn-on threshold and increased stability, reliability, and current density, which will expand the range of their application.”
- Add a schematic for the synthesis process (line 67)
Authors’ response:
The scheme of the synthesis process consists of only two iterations (CVD synthesis of CNT arrays and plasma treatment). In this case, we believe that an additional scheme is unneeded and would be superfluous.
- Check (line 71, 82)
Authors’ response:
The sign is corrected.
- not very clear. (line 119)
Authors’ response:
This paragraph has been improved.
These points are clarified in the manuscript as: “The positions and the number of light-colored layers formed on the side of the plasma treated sample roughly correspond to those in the layers of the initial sample enriched with iron nanoparticles. These layers exhibit uneven lateral plasma etching (Figure 1b) ”
- petals (line 160)
Authors’ response:
In the manuscript, we use the word "petals" to refer to graphene flakes as they resemble "petals" in appearance.
- any justification, may cite from literature (line 204)
Authors’ response:
Link added to the manuscript: “This change may be caused by the changes in the orientation of graphene layers with respect to the incident X-ray beam [61] ”
- rephrase, unclear what authors are trying to infer.
Authors’ response:
In the scientific literature, the value of the work function of unmodified multi-walled CNTs is of the order of ~4.9 eV. The relevant links have been added to the work: “It is commonly thought that the work function in multi-walled CNTs is in interval 4.7-4.9 eV [71, 72].”
Reviewer 2 Report
this manuscript presents the surface modification of the electrodes of a FE device using the SWCNT preparation through H-Plasma treatment. this is of interest to the readers of this field and is contributing to the field. However, the few comments and questions are essentially due before we proceed with this submission:- the ion migration has been assigned to carrier transport in fig. 6.
- please explain how the cooling can create holes on the surface of the balls?
- the work function reduction of the CNT by doping is possible. However, the fig. 6 cannot describe this individually. heat treatment is required to determine this. is it possible for the authors to perform some heat treatment? the fowler-Nordheim mechanism is also able to talk on this but the authors did not explain it well here.
- the carrier transport and the role of electric field requires citation to DOI: 10.1142/S0217984916500445
- in fig. 4, After hydrogen plasma treatment, the intensity of the iron line increased by a factor of 20. this is huge. how this can be explained? is it by the out-diffusion of the iron to surface?
- the high iron intensity was related to CNT damage in fig. 4. in this case, how the authors relate the better carrier transport of the treated samples to CNT doping?
Author Response
Reviewer #2: Recommendation:
This manuscript presents the surface modification of the electrodes of a FE device using the SWCNT preparation through H-Plasma treatment. this is of interest to the readers of this field and is contributing to the field. However, the few comments and questions are essentially due before we proceed with this submission:
Comments
- the ion migration has been assigned to carrier transport in fig. 6.
Authors’ response:
Of course, ion migration occurs, but it makes a negligible contribution to the value of the field emission current.
- please explain how the cooling can create holes on the surface of the balls?
Authors’ response:
We do not know how exactly these periodic structures are formed on the surface of the balls. We intend to conduct an additional research on these structures and publish a new work on this topic. As a guess, we can state the following:
When a metal ball is cooled down, α-iron crystallizes with a body-centered cubic lattice. Such a crystal will have a stepped surface structure (crystalline faces (110), (120) and others are formed). When thin graphite layers are formed on such a surface, periodic depressions appear.
- the work function reduction of the CNT by doping is possible. However, the fig. 6 cannot describe this individually. heat treatment is required to determine this. is it possible for the authors to perform some heat treatment? the fowler-Nordheim mechanism is also able to talk on this but the authors did not explain it well here.
Authors’ response:
An accurate confirmation and determination of the degree of work function decrease due to alloying requires an additional research, including heat treatment. Perhaps this will be done in further research. However, in this manuscript, we consider the totality of the factors affecting field emission and propose a technique that allows modifying CNT arrays to obtain more efficient field emission cathodes on their basis.
- the carrier transport and the role of electric field requires citation to DOI: 10.1142/S0217984916500445
Authors’ response:
The issues you refer to are interesting, but they are very far indeed from the main idea of our manuscript.
- in fig. 4, After hydrogen plasma treatment, the intensity of the iron line increased by a factor of 20. this is huge. how this can be explained? is it by the out-diffusion of the iron to surface?
Authors’ response:
Yes, quite right, a significant increase in the iron concentration is associated with the release of iron nanoparticles to the surface from the inner regions of carbon nanotubes.
- the high iron intensity was related to CNT damage in fig. 4. in this case, how the authors relate the better carrier transport of the treated samples to CNT doping?
Authors’ response:
The manuscript describes that the improvement in the field emission characteristics can be associated both with an increase in the field enhancement factor upon deformation of the ends of CNTs and with a decrease in the work function due to edge effects on graphene nanoparticles. An increase in the concentration of iron on the surface has no significant effect on the field emission current.
Reviewer 3 Report
To authors
General comments
This manuscript deals with aligned multi-walled carbon nanotube arrays treated with hydrogen plasma for improvement of field emission properties desired for the construction of low-voltage emission cathodes. I perceive this article interesting and well-elaborated. The authors prepared the carbon nanotubes on a boron-doped silicon wafers by CCVD method. Subsequently, the CNT were treated with hydrogen plasma using various microwave generator power. The final CNTs were thoroughly characterized by Raman spectroscopy, SEM, TEM, XPS and NEXAFS. It was shown that the plasma treatment influences the chemical composition of the CNTs and enhances the field emission properties thereof. The manuscript was prepared using good language; its overall layout is correct. As a well-composed article, being a coherent whole, it may be read with interest. Thus, the reviewed article may be a value-added reference for the scientists focused on the improvement of CNTs properties. The subject of the work matches the scope of the Materials journal, therefore I suggest the acceptance of its publication after minor (in fact, minute) revision.
Detailed questions
1/ The Authors state (lines77-80): “The plasma treatment of CNT arrays was conducted in a АХ5250М PECVD reactor (Seki 77 Technotron, Japan) at a high pressure (above 1.5×10^4 Pa) with a power of microwave plasma up to ~5 kW [40]. The working pressure in the chamber was ~ 8×10^3 Pa.” This can be a bit confusing - what was the accurate pressure in the reactor then?
2/ Lines 132-133: “The gas phase of the reactor is supplied by light hydrocarbon molecules such as СН4, С2Н2, С2Н4, С2Н6, etc. [47, 48].” One may infer that the gases were introduced to the reactor, while they are formed by the reaction of CNTs with hydrogen plasma, I suppose?
3/ Lines 143-144: “In the course of cooling, the surface of the balls exhibits a structure of symmetrically arranged holes.” Please, clarify this statement.
4/ The number of self-citations is really high. Of course, this proves the Authors' impressive experience in the subject matter they deal with, but it is pointless to cite works that also cite each other.
Author Response
Reviewer #3: Recommendation:
This manuscript deals with aligned multi-walled carbon nanotube arrays treated with hydrogen plasma for improvement of field emission properties desired for the construction of low-voltage emission cathodes. I perceive this article interesting and well-elaborated. The authors prepared the carbon nanotubes on a boron-doped silicon wafers by CCVD method. Subsequently, the CNT were treated with hydrogen plasma using various microwave generator power. The final CNTs were thoroughly characterized by Raman spectroscopy, SEM, TEM, XPS and NEXAFS. It was shown that the plasma treatment influences the chemical composition of the CNTs and enhances the field emission properties thereof. The manuscript was prepared using good language; its overall layout is correct. As a well-composed article, being a coherent whole, it may be read with interest. Thus, the reviewed article may be a value-added reference for the scientists focused on the improvement of CNTs properties. The subject of the work matches the scope of the Materials journal, therefore I suggest the acceptance of its publication after minor (in fact, minute) revision.
Comments
- The Authors state (lines77-80): “The plasma treatment of CNT arrays was conducted in a АХ5250М PECVD reactor (Seki 77 Technotron, Japan) at a high pressure (above 1.5×10^4 Pa) with a power of microwave plasma up to ~5 kW [40]. The working pressure in the chamber was ~ 8×10^3 Pa.” This can be a bit confusing - what was the accurate pressure in the reactor then?
Authors’ response:
The working pressure in the chamber was ~ 8×10^3 Pa.
In the manuscript, this sentence is corrected to: “The plasma treatment of CNT arrays was conducted in a АХ5250М PECVD reactor (Seki Technotron, Japan) with a power of microwave plasma up to ~5 kW [40].”
- Lines 132-133: “The gas phase of the reactor is supplied by light hydrocarbon molecules such as СН4, С2Н2, С2Н4, С2Н6, etc. [47, 48].” One may infer that the gases were introduced to the reactor, while they are formed by the reaction of CNTs with hydrogen plasma, I suppose?
Authors’ response:
Yes that's right. The following clarifications have been made to the text: “The gas phase of the reactor is supplied by light hydrocarbon molecules such as СН4, С2Н2, С2Н4, С2Н6, etc. formed by the reaction of CNTs with hydrogen plasma [47, 48].”
- Lines 143-144: “In the course of cooling, the surface of the balls exhibits a structure of symmetrically arranged holes.” Please, clarify this statement.
Authors’ response:
We do not know how exactly these periodic structures are formed on the surface of the balls. We intend to conduct an additional research on these structures and publish a new work on this topic. As a guess, we can state the following:
The symmetry of the holes on the surface of the balls is associated with the formation of an iron single crystal and the formation of relatively symmetric faces of the crystal lattice with orientations (110), (120), etc.
- The number of self-citations is really high. Of course, this proves the Authors' impressive experience in the subject matter they deal with, but it is pointless to cite works that also cite each other.
Authors’ response:
The high number of self-citations is associated only with the need to confirm a large number of different aspects and nuances that arise when describing the material obtained.
Reviewer 4 Report
Materials
Manuscript ID928760
“Hydrogen plasma treatment of aligned multi-walled carbon nanotubes arrays for improvement of field emission properties”
The present paper describes the hydrogen plasma modification of CNT arrays for the development of planar low-voltage emission cathodes. The work is well described and the conclusions are supported by the results. So, I wish suggest to publish on Materials the following paper after minor revisions.
#remarks 1
Materials&Methods: The authors should add a list of the materials used providing some data about supplier, purity and so on.
#remarks 2
Plasma treatment: the authors describe that during plasma treatment test, they changed the power of the microwave generator. Is it possible to have some information about the time of exposure and discuss the results obtained not only in terms of the generator power variation but also as time of exposure/generator power variation ratio?
#remarks 3
Figure 2 and 3:the SEM pictures presented are very beautiful. However, is it possible to estimate the diameter of iron sphere and CNT tips reported in Figure 2c and Figure 3a and 3b, respectively?
Author Response
Reviewer #4: Recommendation:
The present paper describes the hydrogen plasma modification of CNT arrays for the development of planar low-voltage emission cathodes. The work is well described and the conclusions are supported by the results. So, I wish suggest to publish on Materials the following paper after minor revisions.
Comments
- Materials&Methods: The authors should add a list of the materials used providing some data about supplier, purity and so on.
Authors’ response:
The list of materials used has been updated: “The CNT arrays were grown on boron-doped KEF 4,5 silicon wafers (orientation [100], size 10×10 mm2) [37, 38] in a horizontal quartz reactor. The reactor with the substrates in its center was evacuated to a pressure of ~10–2 Pa, then filled with an inert gas (Ar, purity 99.999%) to atmospheric pressure, and heated to 800 °C. The reaction mixture (2% ferrocene (purity 99.99%) solution in toluene with a purity of 99.9999%) was injected in the reactor and aerosol was supplied by Ar with a flow rate of 450 mL/min to the synthesis zone.”
- Plasma treatment: the authors describe that during plasma treatment test, they changed the power of the microwave generator. Is it possible to have some information about the time of exposure and discuss the results obtained not only in terms of the generator power variation but also as time of exposure/generator power variation ratio?
Authors’ response:
An increase in the generator power leads to an increase in the etching rate of CNT arrays. Therefore, the processing time was 15 minutes. A significant increase of time leads to complete etching of the CNT array at high power, while a decrease of time will lead to insignificant changes in the structure of the CNT array at low power.
- Figure 2 and 3:the SEM pictures presented are very beautiful. However, is it possible to estimate the diameter of iron sphere and CNT tips reported in Figure 2c and Figure 3a and 3b, respectively?
Authors’ response:
The diameter of the iron balls is about 10 μm, and the diameter of CNT tips is less than 100 nm.